# Optimizing Recombinant Cas9 Expression: Insights from *E. coli* BL21(DE3) Strains for Enhanced Protein Purification and Genome Editing

**DOI:** 10.3390/biomedicines12061226

**Published:** 2024-05-31

**Authors:** Shilpi Agrawal, Made Harumi Padmaswari, Abbey L. Stokes, Daniel Maxenberger, Morgan Reese, Adila Khalil, Christopher E. Nelson

**Affiliations:** 1Department of Biomedical Engineering, University of Arkansas, Fayetteville, AR 72701, USA; mhpadmas@uark.edu (M.H.P.); abbeyb@uark.edu (A.L.S.); damaxenb@uark.edu (D.M.); adilak@uark.edu (A.K.); 2Cell and Molecular Biology, University of Arkansas, Fayetteville, AR 72701, USA; mreese@walton.uark.edu

**Keywords:** SpCas9, protein production, BL21(DE3), CRISPR, genome editing

## Abstract

The CRISPR-Cas9 system is a revolutionary tool in genetic engineering, offering unprecedented precision and efficiency in genome editing. Cas9, an enzyme derived from bacteria, is guided by RNA to edit DNA sequences within cells precisely. However, while CRISPR-Cas9 presents notable benefits and encouraging outcomes as a molecular tool and a potential therapeutic agent, the process of producing and purifying recombinant Cas9 protein remains a formidable hurdle. In this study, we systematically investigated the expression of recombinant SpCas9-His in four distinct *Escherichia coli* (*E. coli*) strains (Rosetta2, BL21(DE3), BL21(DE3)-pLysS, and BL21(DE3)-Star). Through optimization of culture conditions, including temperature and post-induction time, the BL21(DE3)-pLysS strain demonstrated efficient SpCas9 protein expression. This study also presents a detailed protocol for the purification of recombinant SpCas9, along with detailed troubleshooting tips. Results indicate successful SpCas9 protein expression using *E. coli* BL21(DE3)-pLysS at 0.5 mM IPTG concentration. Furthermore, the findings suggest potential avenues for further enhancements, paving the way for large-scale Cas9 production. This research contributes valuable insights into optimizing *E. coli* strains and culture conditions for enhanced Cas9 expression, offering a step forward in the development of efficient genome editing tools and therapeutic proteins.

## 1. Introduction

Overexpression and purification of recombinant proteins have revolutionized protein production by utilizing various expression systems, including bacteria, yeast, insect, or mammalian cells [1]. Traditional methods of protein isolation, such as centrifugation and chromatography, have been supplanted by recombinant DNA technology, where DNA encoding desired proteins is cloned into expression systems for efficient production. However, the purification of the target protein from the proteome is crucial, leading to the widespread use of affinity chromatography, a technique for distinguishing proteins based on specific interactions with the matrix [2,3].

Researchers employ affinity chromatography to facilitate the separation of desired proteins from complex mixtures of proteomes [4]. Affinity tags are attached to the C or N terminus or placed at specific sites within the protein (internal tags) and bind to tag-specific resins during purification, allowing elution of the desired protein [5]. Genetic fusion of affinity tags influences various aspects of the protein, including cellular localization, proper folding, expression level, and solubility. The choice of purification resin is also guided by the affinity tag, impacting protein purity, yield, number of purification steps, and cost [6]. Protein tags, such as the His-tag, Glutathione-S-Transferase (GST) tag, Flag tag, Hemagglutinin (HA) tag, and Maltose Binding Protein (MBP) tag, can enhance protein solubility and influence correct folding [4].

Histidine tags, particularly the 6xHis tag, consist of at least six histidine residues and are widely employed for protein purification. This tag offers several advantages over other tags, such as its small size (0.8 kDa), ease of use, and cost-effectiveness. Its small size reduces the likelihood of altering the target protein’s characteristics [7]. Additionally, incorporating a protease cleavage site between the His-tag and the protein’s N/C terminus allows safe removal of the His-tag post-purification using enzymes like Tobacco Etch Virus (TEV), Factor Xa, or Thrombin [8,9]. The ability of histidine to form bonds with transition metal ions, such as nickel and cobalt, is leveraged in purifying target proteins through immobilized metal affinity chromatography (IMAC). IMAC separates proteins based on their interaction with immobilized metal ions (Co^2+^ and Ni^2+^) on solid chelating resin. Different chelators like nitrilotriacetic acid (NTA) and iminodiacetic acid (IDA) are commonly used, with Co^2+^ providing high-purity preparations and Ni^2+^ offering excellent protein yields. Effective binding of His-tags to IMAC resin occurs in near-neutral buffer conditions, and elution of the His-tagged protein is achieved using high concentrations of imidazole [10].

In the context of gene editing, particularly with Clustered Regularly Interspaced Short Palindromic Repeats (CRISPR), the Cas9 enzyme plays a central role [11]. The CRISPR/Cas system, composed of guide RNA (gRNA) and Cas9, facilitates precise genetic manipulation by inducing double-strand breaks at specific genomic loci [12]. Despite the prominence of Cas9, other Cas proteins, such as Cas12, Cas13, and Cas3, offer diverse functionalities, expanding the applications of CRISPR in gene therapy and molecular research [13]. *Streptococcus pyogenes* Cas9 (SpCas9) is the most widely used Cas9 enzyme, and its production and purification, typically from *Escherichia coli*, present challenges despite ongoing efforts to simplify the process [14]. The scalability and cost-effectiveness of Cas RNP production remain challenging [15,16].

As the focus on laboratory-scale production of Cas9 proteins, especially SpCas9, increases, there is a need for addressing challenges and optimizing small-scale production. However, the existing literature lacks comprehensive studies on large-scale production challenges. Laboratory-scale production serves as a crucial intermediary between exploratory protein purification and industrial-scale processes. Despite the suitability of *E. coli* for heterologous expression, challenges such as the large size of SpCas9 and the presence of rare codons may impact overall yield. Existing protocols lack standardization and vary in bacterial strains, culture conditions, and expression vectors [17,18]. The optimization process involves evaluating competent cell strains, IPTG concentrations, post-induction times, and temperatures to ensure scalability for industrial production [19,20]. The significance of these efforts lies in bridging the gap between exploratory protein purification and large-scale industrial processes [21,22].

Here, we have utilized BL21(DE3), BL21(DE3)-Star, BL21(DE3)-pLysS, and Rosetta2 *E. coli* strains. However, each of these competent cell strains has inherent benefits due to their makeup, causing some to be more efficient in protein expression than others. One of the simplest strains of BL21, BL21(DE3), includes a T7 RNA polymerase gene, which means that it can express proteins with the T7 promoter upstream in its genes, unlike regular BL21 which only has the *E. coli* RNA polymerase, making it specific to proteins with *E. coli* promoters. The limitation of this strain comes, not from its inability to express, but rather its likelihood to have leaky expression [23]. That, however, is one of the reasons for the development of BL21(DE3)-pLysS. BL21(DE3)-pLysS has a similar makeup to BL21(DE3), but contains a T7 lysosome expressed in low levels to decompose the T7 polymerase before induction while not hindering T7 polymerase expression post induction, but also reducing the basal level expression of the gene of interest, thus making it suitable for the expression of toxic genes [24]. BL21(DE3)-Star is another strain aimed at improving the BL21(DE3) strain. BL21-Star contains a mutated variant of the RNAse gene rne131. This mutation truncates the C-terminal of the RNAse E enzyme produced by rne131 which plays a role in mRNA degradation. This then increases mRNA stability and overall efficiency for the production of heterologous proteins. However, due to similar issues as BL21(DE3) and the lack of the T7 lysosome, it is still not suitable for expression of toxic genes. Rosetta2 is another strain with more specific uses. Once again, it shares the inclusion of a T7 polymerase with the other strains; however, it contains a plasmid containing tRNAs designed to enhance expression of rare codons in *E. coli*. While this strain does have enhanced expression for eukaryotic proteins containing these codons, otherwise it falls behind the other strains [23,24].

Here, we introduce a novel approach to address the challenges associated with synthesizing a functionally operational SpCas9-His protein. By systematically investigating SpCas9-His expression across various *E. coli* strains (Figure 1) and optimizing culture conditions, including temperature and IPTG concentration, this study identifies BL21(DE3)-pLysS as the most efficient strain for SpCas9 production (Table 1). Additionally, a detailed purification protocol and troubleshooting tips are presented, offering insights into enhancing Cas9 expression efficiency (Figure 2). These findings contribute to the development of more efficient genome editing tools and therapeutic proteins and pave the way for large-scale production of SpCas9 in the laboratory, addressing a critical need in the field.

## 2. Materials

Ensure all solutions are prepared with ultrapure (deionized) water to minimize contaminants that may interfere with experimental outcomes. Use analytical-grade reagents for the preparation of solutions to ensure high purity and consistency. Unless otherwise specified, store all reagents and solutions at room temperature to maintain stability and prevent alterations in chemical composition.

### 2.1. Biological Materials

Plasmid—pET-28b-Cas9-His (Addgene, Watertown, MA, USA, cat. no. 47327)

### 2.2. Competent Cells

BL21(DE3)-pLysS (Millipore Sigma, Burlington, MA, USA, cat. no. 69451-3)BL21(DE3)-Star (ThermoFisher, Waltham, MA, USA, cat. no. C601003)Rosetta2 (Millipore Sigma, cat. no. 71402-3)BL21(DE3) cells (Millipore Sigma, cat. no. 69450-3)

### 2.3. Reagents

Tris-HCl (VWR, Radnor, PA, USA, cat. no. 71003-494)KCl (VWR, cat. no. 71003-522)Imidazole (MP Biomedicals, Santa Ana, CA, USA, SKU: 02102033-CF)Glycerol (Fisher Scientific, Hampton, NH, USA, cat. no. BP229-4)HEPES (VWR, cat. no. VWRB30487)DTT (VWR, cat. no. 97061-338)LB Broth, Lennox (Fisher Scientific, cat. no. BP1427-2)LB Agar, Miller (Fisher Scientific, cat. no. BP1425-500)Kanamycin Sulphate (VWR, cat. no. 75856-686)Chloramphenicol (VWR, cat. no. TCC2255)IPTG (ThermoFisher, cat. no. 15529019)SDS (Biorad, Hercules, CA, USA, cat. no. 1610301)30% Acrylamide solution (Biorad, cat. no. 1610156)Ammonium persulfate (Biorad, cat. no. 1610700)TEMED (Biorad, cat. no. 1610801)Phenylmethylsulfonyl fluoride (PMSF) (Millipore Sigma, cat. no. 10837091001)Methanol (VWR, cat. no. EM-MX0475-1)Ethanol (VWR, cat. no. 89125-188)Acetic Acid (VWR, cat. no. BDH3094)HisPrep FF 16/10 (Cytiva, Marlborough, MA, USA, cat. no. 28936551)Deionized water (facilitated by the University of Arkansas)Pierce Bradford protein assay kit (ThermoFisher, cat. no. 23200)Qiagen QIAprep Spin Miniprep Kit (Qiagen, Germantown, MD, USA, cat. no. 27104)2x Laemmli Sample Buffer (Biorad, cat. no. 1610737)β-Mercaptoethanol (BME) (Millipore Sigma, cat. no. 444203)SDS-PAGE Protein ladder (GenScript, Piscataway, NJ, USA, cat. no. M00624)Western Blot Protein Ladder (Biorad, cat. no. 1610373)Cas9 Monoclonal Antibody (Thermofisher, cat. no. MA1-201)Rabbit anti-mouse IgG Secondary Antibody, HRP (ThermoFisher, cat. no. 61-6520)Tris base (Genesee Scientific, El Cajon, CA, USA, cat. no. 18-144)EDTA-free Protease Inhibitor cocktail (Roche, Indianapolis, IN, USA, cat. no. 11873580001)lysozyme (Millipore Sigma, cat. no. L6876)Clarity Western ECL Substrate (Biorad, cat. no. 1705061)NaCl (VWR, cat. no. EM7760)Tween 20 (Millipore Sigma, cat. no. P9416)Great Value Instant Nonfat Dry Milk (Walmart, Bentonville, AR, USA)DNeasy Blood & Tissue Kit (Qiagen, cat. no. 69504)Q5^®^ High-Fidelity DNA Polymerase (New England Biolabs, Ipswich, MA, USA, cat. no. M0491)10 X NEBuffer™ r3.1 (New England Biolabs, cat. no. B6003)QIAquick PCR purification (Qiagen, cat. no. 28104)Proteinase K (New England Biolabs, cat. no. P8107S)Agarose (Genesee Scientific, cat. no. 20-102GP)1 kB Plus DNA Ladder (New England Biolabs, cat. no. N3200L)Glacial Acetic Acid (Millipore Sigma, cat. no. PHR1748)EDTA, Disodium Salt, Dihydrate, Crystal (Avantor, Radnor, PA, USA, cat. no. 4040-04)Coomasie blue stain (Biorad, cat. no. 1610803)10 X SDS-PAGE running buffer (ThermoFisher, cat. no. NP0001)Trans-Blot Turbo transfer packs (Biorad, cat. no. 1704159EDU)Out glass plt w/1mm (Biorad, cat. no. 1653311)Inner glass plate (Biorad, cat. no. 1653308)Mini-PROTEAN Casting Stand (Biorad, cat. no. 1653303)Casting frame (Biorad, cat. no. 1653304)Mini-PROTEAN Comb 10W 1.0 mm 44 µL (Biorad, cat. no. 1653311)Roller (Biorad, cat. no. 1651279)Gel releasers (Biorad, cat. no. 1653320)

### 2.4. Equipment

1 L and 500 mL Erlenmeyer Flasks (Chemglass Life Sciences, Vineland, NJ, USA)Temperature-Controlled Shaking Incubator (New Brunswick Scientific, Edison, NJ, USA)Temperature-controlled incubator (Fisher Scientific, cat. no. 13-762-721)Mechanical Device to Disrupt *E. coli* Cells:Options: Ultrasonicator (Qsonica Sonicators, Newtown, CT, USA, cat. no. Q55-110), French press, or cell homogenizer.BioTek Synergy LX Multi-Mode Reader (Agilent, Santa Clara, CA, USA)NanoDrop One to check the OD (Thermofisher Scientific, cat. no. ND-ONE-W)Refrigerated Centrifuge and Centrifuge Bottles (500 mL and 50 mL) (Eppendorf, Enfield, CT, USA)AKTA Start for protein purification (Cytiva, cat. no. 29022094)Oakridge Tubes (Thermofisher Scientific)Both 50 and 15 mL falcon tubes (Genessee Scientific, cat. no. 21-106 & 21-103)Ultrafiltration Centrifugal Concentrating Devices (Centricon) with Appropriate Molecular-Weight Cut-Offs (Sigma Aldrich, St. Louis, MO, USA)ChemiDoc Imaging System (Biorad, cat. no. 12003154)Trans-Blot Turbo Transfer System (Biorad, cat. no. 1704150)Mini-Sub Cell GT (Biorad, cat. no. 1664000EDU)Mini Rocker (Benchmark Scientific, Sayreville, NJ, USA, cat. no. BR1000)Mini-Protean Tetra Vertical Electrophoresis Cell (Biorad, cat. no. 1658001FC)

### 2.5. Solutions

Kanamycin solution (50 mg/mL)—Dissolve 0.5 g of Kanamycin sulfate in 10 mL of deionized water, vortex to ensure complete solubility, sterilize by filtering through a 0.2 µm filter, aliquot, and store at −20 °C for up to six months. Add 1 mL of kanamycin stock (at 50 mg/mL) per liter of LB media to obtain a final concentration of 50 µg/mL.Chloramphenicol solution (30 mg/mL)—Dissolve 0.34 g of chloramphenicol in 10 mL of 100% ethanol, vortex to ensure complete solubility, sterilize by filtering through a 0.2 µm filter, aliquot, and store at −20 °C for up to six months. Add 1 mL of chloramphenicol stock (at 30 mg/mL) per liter of LB media to obtain a final concentration of 30 µg/mL.IPTG solution (1 M)—Weigh 2.38 g IPTG, dissolve in 8 mL deionized water, adjust to a final volume of 10 mL, sterile filter using a 0.22 µm syringe filter, aliquot the stock, and store at −20 °C for up to one year.Staining solution (400 mL)—Prepare a staining solution by combining 180 mL of deionized water, 180 mL of methanol, 40 mL of acetic acid, and 7 g of Coomassie Blue, ensuring thorough mixing.Destaining solution (1 L)—Create a destaining solution by mixing 400 mL of methanol and 200 mL of acetic acid, and filling the container to 1 L with deionized water, followed by thorough mixing.Loading dye solution—Add 50 µL of BME to 950 µL of 2× Laemmli Sample BufferLB agar—Combine 20 g of LB agar powder with 500 mL of deionized water, autoclave, cool, and pour into Petri dishes.LB media/broth—Mix 10 g of LB broth in 500 mL of deionized water, autoclave, and store at 4 °C.PMSF (Phenyl Methylsulfonylfluoride) solution: Concentration: 34.8 mg/mL in 100% ethanol (Storage: −20 °C)Glycerol stock—Mix an overnight bacterial culture with 50% glycerol at a 1:1 ratio, aliquot into cryovials, and freeze at −80 °C for long-term storage as glycerol stocks.

### 2.6. Buffers

Lysis Buffer: Composition: 20 mM Tris-HCl pH 8.0, 250 mM KCl, 20 mM imidazole, 10% glycerolWash Buffer: Composition: 20 mM Tris-HCl pH 8.0, 800 mM KCl, 30 mM imidazole, 10% glycerolElution Buffer: Composition: 20 mM HEPES pH 8.0, 500 mM KCl, 250 mM imidazole, 10% glycerolExchange Buffer: Composition: 20 mM HEPES pH 7.5, 500 mM KCl, 20% glycerol, 1 mM DTTNote: Add DTT just before dialysis. For concentrating, add DTT to the fresh Exchange Buffer the next day.SDS-PAGE Running Buffer: Prepare 1 X running buffer by diluting the 10 X buffer to 1 L. Add 100 mL of 10× commercially available running buffer to 900 mL of deionized water.10 X TBS: Composition: 12 g Tris Base, 44 g NaCl, set to pH 7.61 X TBST: Composition: 100 mL of 10 X TBS, 500 μL Tween20, make volume up to 1 LBlocking Buffer: Composition: 2.5 g skim milk, 50 mL 1 X TBST0.5 M EDTA (pH 8.0): Composition: Add 186.1 g of EDTA in 800 mL, adjust pH and then make up to 1 L.50 X TAE: Composition: 243.3 g Tris Base, 57.2 mL glacial acetic acid, 100 mL 0.5 M EDTA (pH 8.0), and make up volume to 1 L.

## 3. Procedure

The recombinant SpCas9-His protein was produced using the pET-28b-Cas9-His plasmid acquired from Addgene (cat. no. 47327). This plasmid facilitates the expression of SpCas9 fused in-frame with a C-terminal nuclear localization signal (NLS) sequence and 6 histidine amino acids. The comprehensive procedure and timeline, starting from isolating the plasmid from the agar stab (supplied by Addgene) to the purification and characterization of SpCas9-His, is detailed in Table 2.

### 3.1. Preparation of Competent Cells

Autoclave the following materials: 500 mL LB media, 2500 mL centrifuge bottles, Eppendorf tubes, pipette tips (if needed), 0.1 M CaCl_2_, and 0.1 M CaCl_2_ in 30% glycerol.Inoculate 10 mL of LB media with BL21-pLysS (or any desired competent cell) and grow for 12–16 h on shaker at 250 rpm.Take 1% of previous culture (e.g., 100 μL of 10 mL) in 500 mL LB media and grow the competent cells on the shaker at 250 rpm until optical density (OD) is between 0.4–0.5. Transfer to the autoclaved centrifuge bottles and centrifuge at 4500 rpm for 15 min at 4 °C.Remove the supernatant and resuspend the pellet in 1 mL of 0.1 M CaCl_2_, mixing carefully on ice. Incubate on ice for 15 min. Then, centrifuge for 4500 rpm for 15 min at 4 °C and discard the supernatant.Prechill 0.1 M CaCl_2_ in 30% glycerol and Eppendorf tubes and then resuspend the pellet in 1 mL of 0.1 M CaCl_2_ in 30% glycerol and mix carefully on ice.Aliquot 50 μL in prechilled Eppendorf tubes and store all tubes at −80 °C.

### 3.2. Preparation of Plasmid from Addgene

Spread a small amount of the agar stab from Addgene on an LB agar plate with kanamycin (1 μL per 1 mL). Leave for 12–16 h at 37 °C.Next day, isolate 1 colony and put in 10 mL of LB media and 10 μL of kanamycin and leave for 12–16 h at 37 °C.To create a glycerol stock, combine and gently mix 500 μL from the bacterial culture and 500 μL of 50% glycerol (1:1 ratio) and store at −80 °C. The SpCas9-His plasmid is isolated from the remaining bacterial culture using the QIAprep Spin Miniprep Kit protocol.

### 3.3. Bacterial Transformation

For transformation, if competent cells are frozen at −80 °C, thaw on ice for ~15 min.Add 5 μL of His-Cas9 plasmid (concentration—5 ng/μL) and 50 μL of competent cells to a 1.5 mL microcentrifuge tube and incubate on ice for 30 min.Note—For transformation to *E. coli* competent cells, use 10–50 ng of the Cas9 plasmid, and the maximum volume that can be used in the transformation is about 10% of the total volume.Heat shock at 42 °C for 45 s and then immediately put it on ice for 3 min. Add 250 μL of SOC media and incubate on shaker for 1 h at 37 °C.Plate 50–100 μL on an agar plate with kanamycin (50 µg/mL) for BL21(DE3), BL21(DE3)-pLysS, and BL21(DE3)-Star, and Chloramphenicol (30 µg/mL) and kanamycin (50 µg/mL) for Rosetta2 as the antibiotic and incubate at 37 °C in an incubator for 12–16 h.After incubation time, take 1 colony from the plate and use it to inoculate in 10 mL of LB media and 10 μL of antibiotic (kanamycin for all the 4 competent cells and kanamycin and chloramphenicol for Rosetta2) (1 μL for every 1 mL of LB media) and incubate for 12–16 h.To create a glycerol stock, combine and gently mix 500 μL from each culture (BL21(DE3), BL21(DE3)-pLysS, BL21(DE3)Star, and Rosetta2) and 500 μL of 50% glycerol (1:1 ratio) and store at −80 °C.

### 3.4. Small-Scale Expression of Recombinant SpCas9-His Protein

#### 3.4.1. Preparation of Starter Culture

In a 50 mL centrifuge tube, add 10 mL of autoclaved LB broth and 10 µL of kanamycin (50 μg/mL) for BL21(DE3), BL21(DE3)-pLysS, and BL21(DE3)Star, and chloramphenicol (30 µg/mL) and kanamycin (50 µg/mL) for Rosetta2.Transfer 5 µL from the His-Cas9 glycerol stocks into the LB broth with the appropriate antibiotic.Incubate the bacterial culture on a shaker (250 rpm) at 37 °C overnight (12–15 h).

#### 3.4.2. Preparation of Small-Scale Bacterial Culture for Protein Overexpression

In a 250 mL flask, prepare 50 mL of sterile LB broth. Inoculate with 2.5 mL (5%) of the starter culture and the appropriate antibiotic concentration (50 µL).Grow the cells at 37 °C and 250 rpm until the optical density at 600 nm reaches ~0.4 to 0.6.Extract 1 mL of bacterial cells from the flask as a pre-induction (un-induced) sample for SDS-PAGE analysis.Induce the culture for SpCas9-His overexpression by adding 50 μL of 1 M IPTG stock solution, achieving a final concentration of 1 mM. Incubate the culture at 30 °C on the shaker.Induction typically takes 5–6 h. Following induction, centrifuge the remaining bacterial cell culture in 50 mL centrifuge bottles at 6000 rpm (10,620× *g*) for 20 min at 4 °C. Discard the supernatant without disturbing the bacterial cell pellet.Resuspend the cells (in 1 mL of freshly prepared lysis buffer) and centrifuge again at 6000 rpm (10,620× *g*) for 15 min at 4 °C.Discard the clear supernatant, resuspend the cells in lysis buffer and load it as a post-induction (induced) sample for subsequent SDS-PAGE analysis, looking for a distinct band indicating the molecular size of SpCas9-His (~160 kDa).

### 3.5. Large-Scale Expression of Recombinant SpCas9-His Protein

#### 3.5.1. Preparation of Starter Culture

Sterilize a flask containing 50–100 mL of LB broth by autoclaving. Then, add kanamycin (50 μg/mL) to the sterile LB media.Add 100 µL of the glycerol stock of SpCas9-His into the LB broth with kanamycin.Incubate the bacterial culture on a shaker (250 rpm) at 37 °C for 12–15 h.

#### 3.5.2. Preparation of Large-Scale Bacterial Culture for Protein Overexpression

In a 1 L Erlenmeyer flask, prepare 400 mL of sterile LB broth. Inoculate with 20 mL (5%) of the starter culture and the appropriate antibiotic concentration.Grow the cells under consistent conditions at 37 °C and 250 rpm until the optical density at 600 nm reaches ~0.4 to 0.6.Induce the culture to overexpress SpCas9-His by adding 40 μL of 1 M IPTG stock solution, achieving a final concentration of 0.5 mM. Incubate the culture at 30 °C.Induction typically takes 5–6 h. Following induction, centrifuge the bacterial cell culture in 500 mL centrifuge bottles at 6000 rpm for 30 min at 4 °C. Discard the supernatant without disturbing the bacterial cell pellet.Transfer resuspended cells (in 50 mL of freshly prepared lysis buffer) to a new falcon tube (50 mL) and centrifuge again at 6000 rpm (10,620× *g*) for 20 min at 4 °C.Discard the clear supernatant, freeze the cell pellet at −80 °C for future use, and thaw when needed at room temperature before placing it on ice for further application.

### 3.6. Purification Procedure for Recombinant SpCas9-His Protein

#### 3.6.1. Bacterial Cell Lysis

Note—Prior to resuspension of the cell pellet in lysis buffer, add and dissolve Complete EDTA-free Protease Inhibitor cocktail–1 tablet per 50 mL lysis buffer, 1 mM PMSF, and lysozyme to a concentration of 1 mg/mL.

Begin by thawing the cell pellet obtained from the freezer. Add 15 mL of lysis buffer and allow it to thaw for 10 min at room temperature. Subsequently, suspend the cell pellet until the solution becomes turbid. Keep the cell suspension on an ice bath.Employ ultrasonication with an amplitude of 20 W output, applying alternate cycles of 20 s ON and OFF for efficient cell lysis.To further enhance cell lysis, subject the post-sonicated sample to a high-pressure French press homogenizer, applying pressure at 30,000 psi (pounds per square inch) for 3 cycles to extract intracellular components efficiently.

#### 3.6.2. Purification Using Nickel-Sepharose Affinity Chromatography (Figure 3)

Centrifuge disrupted cell suspension at 12,100 rpm for 30 min at 4 °C. Before loading onto the column, pass the supernatant through a 0.45 mm polyethersulfone membrane to eliminate any residual particulates.Extract a small aliquot from the supernatant for SDS-PAGE analysis.Apply the supernatant onto a Ni-Sepharose resin column pre-equilibrated with lysis buffer. Maintain a consistent flow rate of 1–2 mL/min. Wash the column with the lysis buffer until a stable baseline is achieved.Wash the column with 100 mL of wash buffer or until the A280 readings approach background levels. Save the wash-through.Elute the SpCas9-His protein with 50 mL of elution buffer or until the A280 is close to background. Ensure to zero the NanoDrop/spectrophotometer with the appropriate buffer if the latter method is employed.Identify fractions containing the SpCas9-His protein through SDS-PAGE.Concentrate the protein as desired and perform buffer exchange, using exchange buffer to eliminate residual imidazole through Vivaspin 20–50 kDa MWCO spin filters. Aliquot samples and store at −80 °C.Note—Aim to reduce the imidazole concentration below 3 mM from the original 250 mM concentration in the Elution Buffer. This process may take a significant amount of time, so wait until the concentrate is around 1 to 2 mL before diluting the remaining imidazole with a much smaller volume of Exchange Buffer. Otherwise, the entire concentration process may take more than two cycles.Measure the protein concentration by Bradford assay.Freeze 0.5 mL aliquots for long-term storage at −80 °C.

**Figure 3 biomedicines-12-01226-f003:**
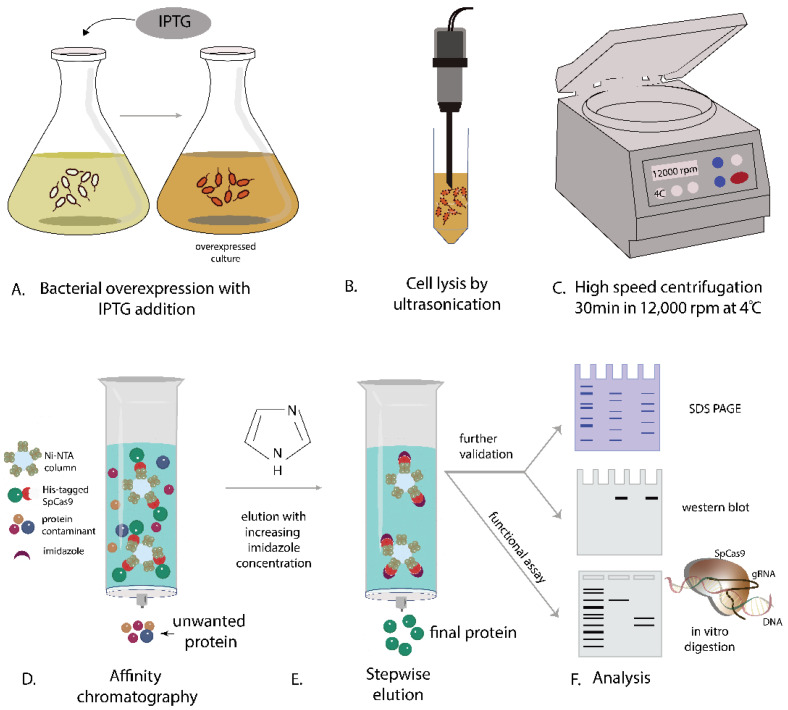
Schematic representation illustrating the sequential steps involved in the purification of SpCas9-His protein using Ni-Sepharose resin and an imidazole gradient.

### 3.7. SDS Page

#### 3.7.1. Preparing the Gel

Press together two glass plates using the casting frame so the indention of the back plate is in between, leaving a slight opening in the plates. Put the casting frame in the casting stand and load deionized water onto the plate opening to the top. Check for leakage.To run SpCas9-His, make 10.5% resolving gel in a 15 mL conical tube. The following volumes coordinate with 1 gel. (Note: 10% APS and TEMED need to be added last because they initiate polymerization)
a.The 10.5% resolving gel is composed of 1.3 mL of deionized water, 1.75 mL of 30% acrylamide, 1.875 mL Tris (Base) at pH = 8.8, 50 μL of 10% SDS, 50 μL of 10% APS, and 10 μL of TEMED.
Pour 10.5% resolving gel into plates and add 1 mL of isopropyl alcohol to eliminate free radicals from the gel. After solidification make a 5% stacking gel solution in a 15 mL conical tube. The following volumes coordinate with 1 gel. (See Note above)
a.The 5% stacking gel is composed of 1.875 mL of deionized water, 0.415 mL of 30% acrylamide, 0.62 mL Tris (Base) at pH = 6.8, 25 μL of 10% SDS, 25 μL of 10% APS, and 10 μL of TEMED.
Pour 5% stacking gel into plates after removing isopropyl alcohol and insert the comb. If gel is not immediately used, wrap gels including the comb in a wet paper towel and foil and store at 4 °C. 

#### 3.7.2. Running Gel

Put the gel into the Mini-PROTEAN Tetra Companion Running Module and put into the buffer tank.Load the buffer tank with SDS-PAGE running buffer until the designated lines. Ensure the module is full regardless of the level of buffer in the buffer tank.Load samples into wells along with ladder and run for approximately 60 min at 200 V and 100 A.

### 3.8. Protein Storage

After the gel is run and pure protein is observed, quantify the protein concentration using a Bradford assay.Store the protein at a final concentration~1 mg/mL (usually the protein concentration is higher than 1 mg/mL after buffer exchange). Prepare single use aliquots and store at −80 °C. The protein remains active for over a year.

### 3.9. Western Blot

During SDS-PAGE, prepare 10 X TBS and 1 X TBST buffers (See Section 2.6 for recipe).After SDS-PAGE is completed, remove the back plate and cut off the stacking gel.Using the Trans-Blot Turbo System, put the bottom blot of the Midi 0.2 μm Nitrocellulose Transfer Packs in one of the compartments. (Note: Be very careful not to touch the blot’s surface while taking out of the package and handling. Additionally, make sure hands and all instruments are coated with deionized water to reduce possible contamination).Add your gel and then the top blot, ensuring each layer is flush through rolling. Run Trans-Blot Turbo on High Molecular Weight.Prepare 50 mL of blocking buffer (See Section 2.6 for recipe) in a 50 mL conical tube.Transfer the bottom blot into box and pour in approximately 15 mL of blocking buffer and incubate on a rocker for 1 h at RT.Dilute the primary antibody (1:1000) in blocking buffer to a final volume of 15 mL and incubate on the rocker for 1 h at RT or overnight at 4 °C.Rinse the blot for 5 min 5 times with 1 X TBST, not directly pouring on the plot.Prepare an additional 15 mL of blocking buffer. Then dilute the secondary antibody (1:10,000) in blocking buffer to a final volume of 15 mL and incubate on the rocker for 1 h at RT. [Note: secondary antibody is conjugated to horseradish peroxidase (HRP)]Rinse the blot for 5 min 6 times with 1 X TBST, not directly pouring on the plot.Prepare the Clarity Max Western ECL Substrate Mixture by combining 6 mL of Peroxide Reagent and 6 mL of Luminol/Enhancer Reagent in a 15 mL conical tube. Add it to the blot and incubate on the rocker for approximately 2 min.To visualize luminescence, place blot in the ChemiDoc on the platform and roll out to make the blot flush with the platform. If marks are saturated in the image, rinse blot with deionized water to remove ECL mixture.

### 3.10. In Vitro Digestion

#### 3.10.1. The Construction of Guide RNA and Amplification of Target Gene

In this experiment, a previously validated guide RNA (gRNA) targeting the human *CKM* gene is utilized. The detailed protocol for constructing in vitro transcribed gRNA is referenced in Padmaswari, M. H. et al. [25].To begin, the human genome is extracted from the HEK293 cell line. The *CKM* fragment is then amplified using appropriate primers with Q5^®^ High-Fidelity DNA Polymerase (Figure 4).Following amplification, the PCR product is purified using Qiaquick PCR purification. This purified *CKM* fragment serves as the template for subsequent steps in the experiment.

#### 3.10.2. The Incubation of CRISPR Ribonucleoprotein Complex

Prepare a reaction mixture of 27 μL at room temperature in the following sequential order:
**Component****Volume****Final Concentration**10 X NEBuffer^TM^ r3.1 (NEB# B6003)3 μL1 X300 nM gRNA3 μL30 nM1 μM SpCas9-His1 μL30 nMNuclease free water20 μL


The CRISPR ribonucleoprotein (RNP) complex is assembled by incubating the in vitro guide RNA: purified SpCas9-His: DNA fragment at nanoMolar concentration ratios of 30:30:1 and 60:60:1.

2.Pre-incubate the mixture at 25 °C for 10 min.3.Gently add 3 μL of a 30 nM purified DNA fragment into the mixture, achieving a final concentration of 3 nM.4.Thoroughly mix the components using pipette mixing.5.Incubate the mixture at 37 °C for 15–60 min.6.Add 1 μL of Proteinase K to the reaction and ensure thorough mixing.7.Allow the reaction to incubate at room temperature for 10 min.

#### 3.10.3. Fragment Analysis

Prepare a 2% TAE agarose gel.Adjust the DNA ladder concentration into 100 ng/μL and load 1 μL of the DNA ladder.Load 30 μL of the in vitro digestion reaction.Run the gel for 30 min at 100 V.Visualize the fragment in gel imager.

## 4. Anticipated Results

*Escherichia coli* has been extensively employed as a platform for recombinant protein expression, leveraging its well-understood genetics, efficient molecular tools, and rapid cell growth. The SpCas9-His protein, the focus of this study, is a sizable molecule with 1390 amino acids and a molecular weight of 158 kDa, sourced from *Streptococcus pyogenes*, making *E. coli* an apt expression system. However, the SpCas9-His coding sequence comprises numerous codons uncommon in *E. coli*, potentially hindering expression due to the limited availability of corresponding tRNAs, which can reduce protein yield or result in truncated proteins [26]. To address this, the BL21(DE3) Rosetta2 strain, equipped with a pRARE plasmid encoding rare tRNAs, is commonly employed to enhance the success of expressing and purifying recombinant proteins containing rare codons, although the direct impact of tRNA compensation on increased protein production remains not fully established [23].

As per existing literature, optimized *E. coli* strains, such as Rosetta2, have been commonly chosen for SpCas9-His protein expression; however, the reported studies vary significantly in expression conditions, displaying concerns about productivity and scalability. Previous research employed complex culture media, low induction temperatures, and extended induction times, often associated with challenges and low SpCas9-His protein expression levels [21,27,28,29]. Moreover, these studies lacked stringent control of bacterial growth in bioreactors.

In this study, SpCas9-His protein expression was initially assessed using the temperature and IPTG concentration conditions (0.5 mM IPTG at 18 °C overnight) from the existing literature [21]. Consequently, we conducted small-scale expression experiments (in shake flasks with 50 mL of LB media) of SpCas9-His protein with various bacterial strains, including BL21(DE3)-pLysS, BL21(DE3)-Star, Rosetta2, and BL21(DE3), and found that BL21(DE3)-pLysS exhibited superior expression compared to others. Therefore, BL21(DE3)-pLysS was chosen for the initial experiments (Figure 5A). Using BL21(DE3)-pLysS, induction was performed at two different concentrations (0.5 mM IPTG and 1 mM IPTG) at 30 °C for 6 h (Figure 5B). Under these conditions, SpCas9-His expression was found to be comparatively lower when induced with 1 mM IPTG. Therefore, 0.5 mM IPTG concentration was chosen for future studies. The cells were collected after 6 h of induction, as described previously. Next, four different induction temperatures (18, 25, 30, and 37 °C) were evaluated. The cell lysate corresponding to each condition was analyzed by SDS-PAGE and the protein expression was found to be highest when induced at 30 °C. The results are presented in Figure 5C.

Next, we evaluated SpCas9-His protein expression in four different *E. coli* strains–BL21(DE3), BL21(DE3)Star, BL21(DE3)-pLysS, and Rosetta2 host strains in shake flasks with 50 mL of LB media. The induction of protein expression was performed using 0.5 mM IPTG at 30 °C for 6 h. The SDS-PAGE analysis revealed that under these conditions, the protein expression in Rosetta2 was notably lower than in BL21(DE3), BL21(DE3)Star, and BL21(DE3)-pLysS (Figure 6). Conversely, SpCas9-His protein expression in the BL21(DE3)-pLysS strain was easily detectable in crude cell extracts, with substantial purification achieved in a single instance of Ni-Sepharose affinity chromatography. Consequently, for large-scale expression, the choice was made to utilize BL21(DE3)-pLysS with an IPTG concentration of 0.5 mM at a temperature of 30 °C (Figure 7).

The observed decrease in SpCas9-His protein expression in Rosetta2 cells, as compared to the regular BL21(DE3) strain, aligns with findings in the literature. Prior studies by Carmignotto et al. confirmed that Rosetta2 cells, grown with chloramphenicol to maintain the pRARE plasmid, showed significantly lower SpCas9-His expression levels than BL21(DE3) [19]. Control experiments with BL21(DE3) harboring the pACYCDuet-1 plasmid, which also carries a chloramphenicol resistance gene, indicated a 35% reduction in SpCas9-His expression. This reduction, although present, was not as severe as observed in the Rosetta2 strain. The unexpected low expression in Rosetta2 was hypothesized to be linked to the presence of rare codons influencing protein translation kinetics and folding. Additionally, qRT-PCR revealed that the reduced SpCas9-His expression in Rosetta2 was associated with a low level of transcribed SpCas9 mRNA, potentially influenced by the large size of the SpCas9 mRNA and metabolic stress induced by the extra pRARE plasmid [19]. This collective evidence supports our findings and emphasizes the influence of additional plasmids on recombinant protein expression, indicating potential metabolic side effects affecting transcription rates.

The SpCas9-His protein was purified from cell crude extracts using affinity chromatography with a His-tag at the C-terminus, as depicted in Figure 7. The SDS-PAGE analysis of the elution step fractions revealed a distinct 158 kDa band, confirming the presence of the purified SpCas9-His protein. On average, this protocol yields 35–40 mg of purified, active, SpCas9-His protein. The purity addressed through SDS-PAGE is >80%. The Western blot analysis was conducted using a SpCas9 antibody to further validate the expression and purification of Cas9 protein. The antibody specifically targeted SpCas9, allowing for the detection of the protein in the samples. The results of the Western blot provide additional confirmation of the presence and molecular weight of the SpCas9-His protein, offering an additional layer of verification alongside SDS-PAGE. This immunodetection technique is valuable for ensuring the specificity of the expressed protein and verifying its successful purification, reinforcing the overall reliability of the experimental outcomes.

The experiment was conducted in triplicate; the additional gel images below represent the other replicates, with error bars indicating standard error of means.

To evaluate the functionality of purified SpCas9-His, we conducted an initial validation of its cleavage activity through in vitro digestion. The CRISPR ribonucleoprotein (RNP) complex was assembled by incubating the in vitro guide RNA: purified SpCas9-His: DNA fragment at a concentration ratio of 30:30:1 and 60:60:1. The densitometric analysis results reveal that the RNP complex formed by in-house purified SpCas9-His displays superior activity when compared to commercially available SpCas9, effectively cleaving the target DNA (~100%) in the expected sizes of 784 bp, which is visible, and 129 bp, which has a smaller mass and is not easily visible on the gel (Figure 8).

## 5. Notes/Troubleshooting

Troubleshooting in protein expression is an iterative process, demanding adaptability based on specific experimental conditions. Consistent monitoring and protocol adjustments are vital for achieving optimal results.

No Protein Expression: If no protein is observed, it indicates plasmid loss. Retransform BL21 Rosetta2 or other competent cells or use another aliquot from the bacterial stock. Verify the plasmid’s presence in the bacterial stock.Expression Host Choice and Yield Parameters: Selecting an appropriate expression host is critical for cloning. Prokaryotes, favored for their ease of handling, cost-effectiveness, and rapid cell growth, may pose challenges in upscaling for large-scale production. Key parameters influencing expression yields include growth temperature, shaker aeration speed, bacterial generation time, induction time, and IPTG concentration. Each variable requires individual testing and optimization. Determining the strain with the highest SpCas9-His expression rate among *E. coli* competent cells is the initial step, followed by testing various IPTG concentrations and post-induction times and temperatures. This process, evaluated at various volumes, should be compared against existing literature methods for continuous improvement.Histidine-Binding Affinity: The recombinant protein’s binding affinity to heparin is contingent on net charge, influenced by pH. Optimal elution under salt gradient conditions and incubation of the heparin-Sepharose column at 4 °C are recommended for efficient binding.Unexpected Protein Size and Proteolysis: Protease contamination or instability of the target protein can lead to proteolysis. Adding protease inhibitors during purification and adjusting temperature conditions is advised. Ni-Sepharose matrix efficiency relies on proper maintenance and storage in 20% ethanol/lysis buffer with 1 mM sodium azide at 4 °C.Detection Challenges: In cases of low protein expression, Western blotting with anti-SpCas9 or anti-His antibodies are recommended for specific detection, even at low concentrations.Time Considerations: The entire procedure, from transformation to the final purification of the SpCas9 protein, takes approximately 126 h. These optimization steps are crucial for making large-scale production more readily achievable.Poor Solubility of Purified Protein: Incompatibility of the lysis buffer or purification conditions with the protein could lead to inefficient lysis or purification. Optimizing lysis and purification buffers to enhance solubility, and including additives like urea or guanidine hydrochloride, if necessary, is recommended.Loss of Protein Stability: Inadequate storage conditions could result in denaturation during storage. The protein should be stored in a suitable buffer at recommended temperatures.Contaminants in Purified Protein: Incomplete washing or elution steps may lead to cross-contamination during purification. Improving washing steps and ensuring proper elution are essential. Implementing additional purification steps may be needed.

## 6. Patent

C.E.N and M.H.P has submitted an invention disclosure related to genome editing.

## Figures and Tables

**Figure 1 biomedicines-12-01226-f001:**
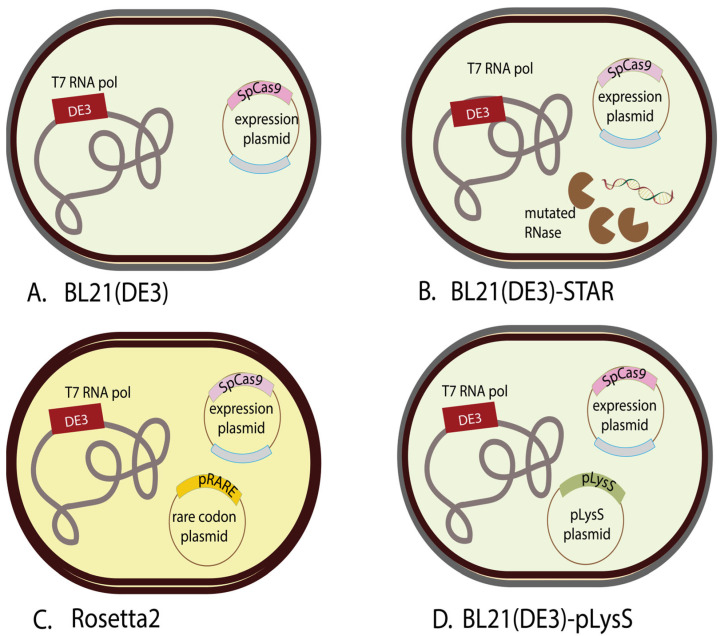
Comparative depiction of competent cells used in this study, including (**A**) BL21(DE3), (**B**) BL21(DE3)-Star, (**C**) Rosetta2, and (**D**) BL21(DE3)-pLysS.

**Figure 2 biomedicines-12-01226-f002:**
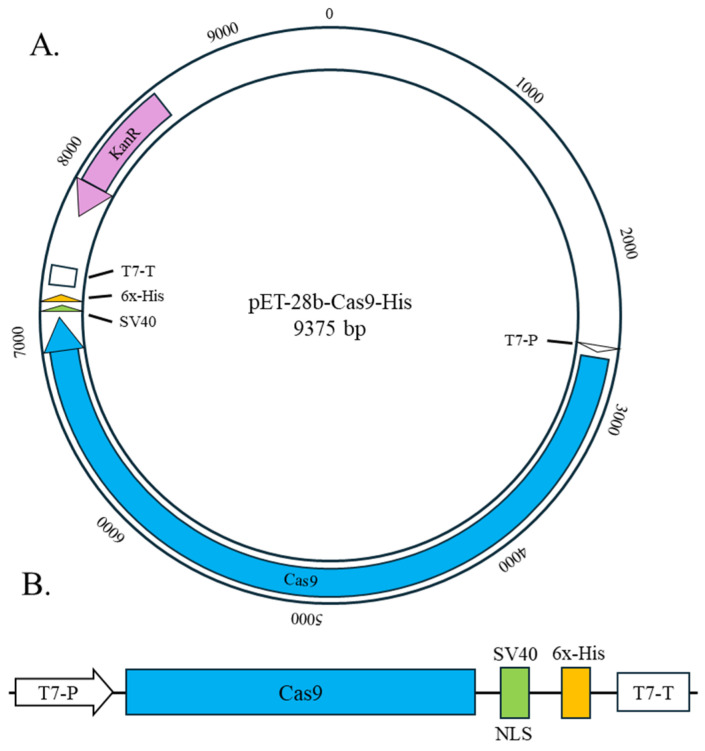
(**A**) Genetic map of pET-28b-Cas9-His obtained from Addgene. (**B**) Transcriptional unit of SpCas9-NLS-6His.

**Figure 4 biomedicines-12-01226-f004:**
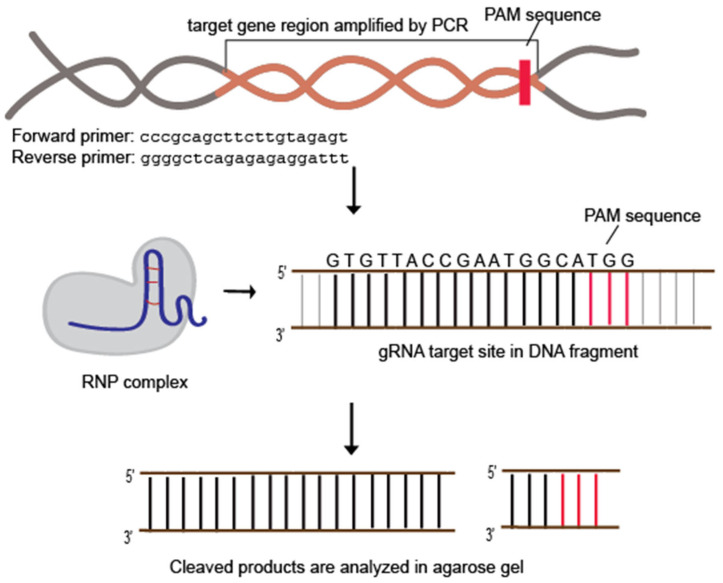
In vitro analysis of RNP (Ribonucleoprotein) complex assembly.

**Figure 5 biomedicines-12-01226-f005:**
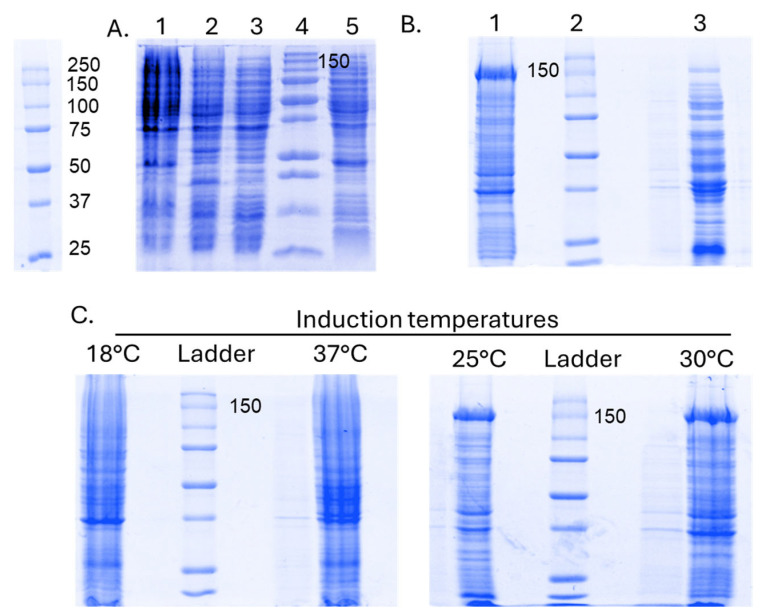
Evaluation of the effects of IPTG concentration and induction temperature on the expression of SpCas9-His protein in BL21(DE3)-pLysS strain of *E. coli*. (**A**) SDS-PAGE analysis of the expression of SpCas9-His protein at 0.5 mM IPTG at 18 °C overnight. (**B**) SDS-PAGE analysis of the expression of SpCas9-His protein at 30 °C but at different IPTG concentrations (lane 1—0.5 mM IPTG, lane 2—protein ladder, lane 3—1 mM IPTG conc); (**C**) Expression of SpCas9-His protein at different induction temperatures (18, 25, 30, and 37 °C).

**Figure 6 biomedicines-12-01226-f006:**
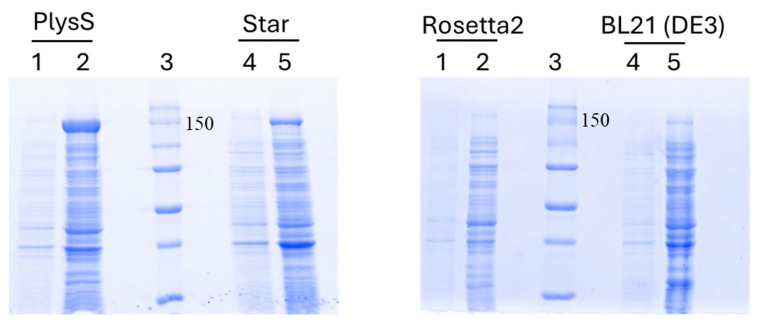
Assessment of SpCas9-His protein expression in BL21(DE3)-pLysS, BL21(DE3)-Star, Rosetta2, and BL21(DE3) cells induced with 0.5 mM IPTG at 30 °C for 6 h in LB medium. Legend—1 and 4: Cell lysate before induction; 2 and 5: Cell lysate post-induction; 3—Protein marker.

**Figure 7 biomedicines-12-01226-f007:**
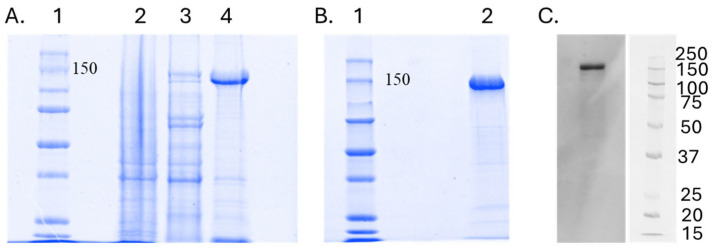
Evaluation of SpCas9-His protein expression in BL21(DE3)-pLysS induced with 0.5 mM IPTG. (**A**) Protein purification profile of SpCas9-His using Ni-Sepharose affinity chromatography. 1: protein ladder; 2—fraction from the wash step; 3—first 5 mL of the fraction from the elution step. This is an important step as it gets rid of the lower contamination and 4—remaining fraction from the elution step. (**B**) SDS PAGE gel after buffer exchange to ensure pure protein. 1: protein ladder; 2—pure protein after buffer exchange. (**C**) Western blot analysis of SpCas9-His expression. Legend: SpCas9-His protein and pre-stained protein MW marker.

**Figure 8 biomedicines-12-01226-f008:**
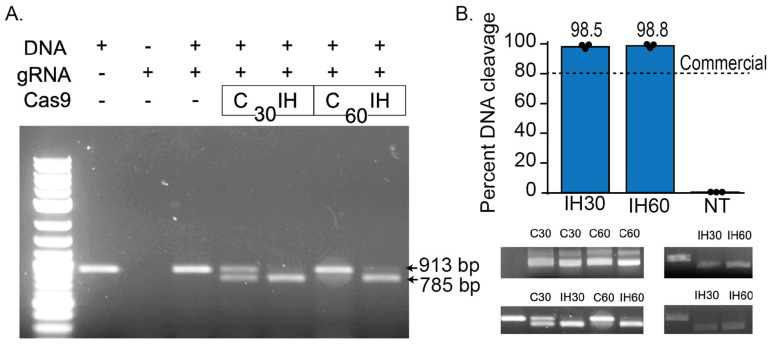
(**A**) In vitro cleavage assay. Only a complex of SpCas9 and gRNA could digest the plasmid DNA. “C” represents commercial Cas9, while “IH” represents in-house Cas9. The numbers 30 and 60 denote the ratio of gRNA:SpCas9:DNA. For instance, 30 corresponds to a ratio of 30:30:1 (nM), and 60 corresponds to a ratio of 60:60:1 (nM). (**B**) The densitometric analysis of in vitro digestion assay of the cleaved DNA (784 bp) band, as monitored by agarose gel electrophoresis.

**Table 1 biomedicines-12-01226-t001:** Summary of the *E. coli* strains used in the current study.

Bacterial Strain	Features	Benefits	Growth Condition	Relative Ranking in Terms of the Protein Yield in This Study
BL21(DE3)	Inclusion of T7 polymerase.	Expression of nontoxic genes.	Kanamycin (50 μg/mL)	3
BL21(DE3)-Star	Includes mutated RNase for increased mRNA stability.	Tight control expression. Expression of both soluble and insoluble proteins.	Kanamycin (50 μg/mL)	2
BL21(DE3)-pLysS	Includes pLysS plasmid expressing T7 lysosome to hinder pre-induction expression.	Expression of toxic genes. Mitigates leaky expression.	Kanamycin (50 μg/mL)	1
Rosetta2	Inclusion of rare codons not generally found in *E. coli*	Expression of heterogenous proteins containing rare codons.	Kanamycin (50 μg/mL) and Chloramphenicol (30 μg/mL)	4

**Table 2 biomedicines-12-01226-t002:** Brief overview of the experimental design.

Process	Hands on Time (h)	Approximate Time Required (h)
Preparation of plasmid from Addgene (bacterial growth + colony pick-up + plasmid isolation)	1	14 + 14 + 1
Bacterial transformation (transformation + colony pick-up + glycerol stock prep)	1.5	2 + 14 + 0.5
Small-scale expression (bacterial growth + induction + harvesting + SDS PAGE)	3	18 + 6 + 1 + 2
Large-scale expression (bacterial growth + induction + harvesting)	4	18 + 6 + 1
Protein purification (bacterial cell lysis + purification + SDS PAGE)	6	0.5 + 5 + 2
Buffer exchange and protein storage	1	6 + 0.5
Western Blot (SDS PAGE + protein transfer and detection)	4	2 + 6
In vitro digestion	2	6
Total	22.5	126 h/6 days

## Data Availability

Data are contained within the article.

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
