# Peer review of "Optimizing Recombinant Cas9 Expression: Insights from E. coli BL21(DE3) Strains for Enhanced Protein Purification and Genome Editing"

_biomedicines, 2024, doi:10.3390/biomedicines12061226_

Round 1
Reviewer 1 Report
Comments and Suggestions for Authors
The innovation of this study is mainly reflected in the following aspects:
1. This study systematically studied the expression of SpCas9-His protein by expressing four different E. coli strains. It was found that the BL21(DE3) pLysS strain expressed SpCas9 protein better. At the same time, this study also provides a detailed SpCas9 protein purification protocol and solutions to difficult problems.
2. Discovered the optimal conditions for BL21(DE3)pLysS strain to express SpCas9 protein at a concentration of 0.5 mM IPTG, This discovery provides a potential way to produce Cas9 proteins on a large scale.
3. This study optimized E. coli strains and culture conditions to improve the expression efficiency of Cas9 protein. This lays the foundation for the development of efficient gene editing tools and therapeutic proteins. In summary, this study provides a method for efficient expression and purification of SpCas9 protein through systematic experiments and optimization, and provides valuable insights for further optimization of Cas9 expression. This is of great significance for promoting the development of gene editing tools and therapeutic proteins.
Existing problems: Figure 8 lacks a description of the size of DNA fragments, and more detailed descriptions of DNA fragment size changes are needed.
Comments on the Quality of English Languageno obvious errors
Author Response
Overview: At the outset, we would like to thank all the reviewers for their thoughtful and constructive comments on our original submission. The reviewers’ suggestions and comments have been extremely useful in improving the quality of our manuscript. We have modified the manuscript to address all the comments which are reproduced in their entirety below. Major changes include additional replicates of the in vitro digestion showing that laboratory produced Cas9 is active. In addition, more discussion and data related to initial optimization was included.
Response: Thank you for the encouraging comments.
Response: We agree with the reviewer that more information on the DNA fragment size should be added to the manuscript. In this context, we have revised the manuscript with the following sentences (Lines 636-641): The densitometric analysis results reveal that the RNP complex formed by in-house purified SpCas9-His displays superior activity when compared to commercially available SpCas9, effectively cleaving the target DNA (~100%) in the expected size, 784 bp and 129 bp (not visible because of the small mass of the fragment) (Figure 8).
Reviewer 2 Report
Comments and Suggestions for Authors
Dear authors,
Congratulations for writing a very detailed and clear paper. I would like to suggest some points of improvement:
Major comment:
1) The robustness of the in vitro cleavage assay is a bit doubtful: How often was this assay performed? The paper would benefit from repeating this experiment at least three times and have a proper quantification with statistical analysis.
2) You describe a very detailed protocol of a basic technique, please highlight the novelty of the described procedure.
Minor points of improvement:
1) Figures are of too low resolution
2) In vitro should be written in italics
Author Response
Overview: At the outset, we would like to thank all the reviewers for their thoughtful and constructive comments on our original submission. The reviewers’ suggestions and comments have been extremely useful in improving the quality of our manuscript. We have modified the manuscript to address all the comments which are reproduced in their entirety below. Major changes include additional replicates of the in vitro digestion showing that laboratory produced Cas9 is active. In addition, more discussion and data related to initial optimization was included.
Dear authors,
Congratulations for writing a very detailed and clear paper. I would like to suggest some points of improvement:
Response: Thank you for the encouraging comment.
Major comment:
1) The robustness of the in vitro cleavage assay is a bit doubtful: How often was this assay performed? The paper would benefit from repeating this experiment at least three times and have a proper quantification with statistical analysis.
Response: In the light of reviewer’s comments, we have included another figure (figure 8B) in the manuscript. In vitro cleavage assay was performed in triplicate and individual data points are provided and compared to no treatment controls and commercial Cas9.
2) You describe a very detailed protocol of a basic technique, please highlight the novelty of the described procedure.
Response: In accordance with the reviewer’s suggestion, we have highlighted the novelty of the described purification protocol in the revised manuscript. The edited part can be found in lines 114-124. “In this research, we introduce a novel approach to addressing the challenges associated with synthesizing a functionally operational SpCas9-His protein (Figure 2). By systematically investigating SpCas9-His expression across various E. coli strains (Figure 1) and optimizing culture conditions, including temperature and different IPTG concentrations, this study identifies BL21 (DE3) PLysS as the most efficient strain for SpCas9 production (Table 1). Additionally, a detailed purification protocol and troubleshooting tips are presented, offering valuable insights into enhancing Cas9 expression efficiency. These findings contribute to the development of more efficient genome editing tools and therapeutic proteins and pave the way for large-scale production of SpCas9 in the laboratory, addressing a critical need in the field.”
Minor points of improvement:
1) Figures are of too low resolution
Response: As per the reviewer’s suggestion, we have uploaded a separate figure file for all the figures.
2) In vitro should be written in italics
Response: We thank the reviewers for pointing out the mistake. We have now modified it in the revised manuscript.
Reviewer 3 Report
Comments and Suggestions for Authors
The article is devoted to the study of such an important process as optimizing of recombinant protein Cas9 expression, correct choice of cell line for protein expression and culture conditions optimization. The subject of the work is relevant for all researchers dealing with gene editing, and especially for those who use RNP in their work. The work is mostly well thought out, implemented and written however there are a few remarks.
Major remarks:
1. There is no explanation why the BL21 (DE3) pLysS line was selected for the first experiments on selecting the cultivation temperature and IPTG concentration.
2. It is logical that if the conditions were selected specifically for the BL21 (DE3) pLysS line, then the expression and production of the protein in this line were the highest. Perhaps other lines simply require different conditions (IPTG concentration and temperature)? It is not clear why this particular line was chosen to start work? Moreover, later in the troubleshooting section the authors write that the line is first determined, and then the conditions are optimized (“Determining the strain with the highest SpCas9-His expression rate among E. coli competent cells is the initial step, followed by testing various IPTG concentrations and post-induction times and temperatures."). This issue is not discussed in the text of the article.
3. There are no conclusions from the results obtained during induction of expression at different cultivation temperatures, there is only a link to the illustration.
Minor:
1. Line 215 - apparently chloramphenicol should be listed here instead of kanamycin.
2. Lines 284 and 298 - Competent cells according to the protocol are suggested to be aliquoted in 100 μl, and later used in 50 μl. Perhaps it would be more convenient to aliquot 50 µL each?
3. Lines 326, 332, 356 - Since the work uses different lines carrying resistance to different antibiotics, it is better to write “LB medium with the appropriate antibiotic” instead of specifically indicating kanamycin.
4. Line 629 – E. coli in italics
Author Response
Overview: At the outset, we would like to thank all the reviewers for their thoughtful and constructive comments on our original submission. The reviewers’ suggestions and comments have been extremely useful in improving the quality of our manuscript. We have modified the manuscript to address all the comments which are reproduced in their entirety below. Major changes include additional replicates of the in vitro digestion showing that laboratory produced Cas9 is active. In addition, more discussion and data related to initial optimization was included.
Reviewer 3:
The article is devoted to the study of such an important process as optimizing of recombinant protein Cas9 expression, correct choice of cell line for protein expression and culture conditions optimization. The subject of the work is relevant for all researchers dealing with gene editing, and especially for those who use RNP in their work. The work is mostly well thought out, implemented and written however there are a few remarks.
Response: Thank you for the encouraging comment.
Major remarks:
- There is no explanation why the BL21 (DE3) pLysS line was selected for the first experiments on selecting the cultivation temperature and IPTG concentration.
Response: In light of the reviewer’s comments, we have included an explanation as to why the BL21 (DE3) pLysS was selected for the initial set of experiments (Lines: 547-561). “In this study, SpCas9-His protein expression was initially assessed using the temperature and IPTG concentration conditions (0.5 mM IPTG at 18°C overnight) from the existing literature21. Consequently, we conducted small-scale expression experiments (in shake flasks with 50 mL of LB media) of SpCas9-His protein with various bacterial strains including, BL21(DE3) pLysS, BL21 (DE3) Star, Rosetta2, and BL21 (DE3) and found that BL21 (DE3) pLysS exhibited superior expression compared to others. Therefore, BL21 (DE3) pLysS was chosen for the initial experiments (Fig. 5A). Using BL21 (DE3) pLysS, induction was performed at two different concentrations (0.5 mM IPTG and 1mM IPTG) at 30°C for 6 hours (Figure 5B). Under these conditions, SpCas9-His expression was found to be comparatively lower when induced with 1mM IPTG. Therefore, 0.5 mM IPTG concentration was chosen for future studies. The cells were collected after 6 h of induction, as described previously. Next, four different induction temperatures (18, 25, 30 and 37 °C) were evaluated. The cell lysate corresponding to each condition was analyzed by SDS-PAGE and the protein expression was found to be highest when induced at 30 °C”. Additionally, figure 5C has been updated accordingly.
- It is logical that if the conditions were selected specifically for the BL21 (DE3) pLysS line, then the expression and production of the protein in this line were the highest. Perhaps other lines simply require different conditions (IPTG concentration and temperature)? It is not clear why this particular line was chosen to start work? Moreover, later in the troubleshooting section the authors write that the line is first determined, and then the conditions are optimized (“Determining the strain with the highest SpCas9-His expression rate among E. colicompetent cells is the initial step, followed by testing various IPTG concentrations and post-induction times and temperatures."). This issue is not discussed in the text of the article.
Response: We thank the reviewers for pointing out the ambiguity. Our paper aims to identify the optimal SpCas9-His expressing bacterial strain. While we acknowledge that further optimization of conditions for different bacterial strains is possible, our intention was to propose a method that is universally applicable and time-efficient. Our current approach achieves peak protein expression within 6 hours post-induction, contrasting with the method outlined in existing literature which requires overnight growth at 18°C. We have now provided a detailed explanation for selecting PLysS for our experiment in the manuscript (lines 547-561).In this study, SpCas9-His protein expression was initially assessed using the temperature and IPTG concentration conditions (0.5 mM IPTG at 18°C overnight) from the existing literature21. Consequently, we conducted small-scale expression experiments (in shake flasks with 50 mL of LB media) of SpCas9-His protein with various bacterial strains including, BL21(DE3) pLysS, BL21 (DE3) Star, Rosetta2, and BL21 (DE3) and found that BL21 (DE3) pLysS exhibited superior expression compared to others. Therefore, BL21 (DE3) pLysS was chosen for the initial experiments (Fig. 5A). Using BL21 (DE3) pLysS, induction was performed at two different concentrations (0.5 mM IPTG and 1mM IPTG) at 30°C for 6 hours (Figure 5B). Under these conditions, SpCas9-His expression was found to be comparatively lower when induced with 1mM IPTG. Therefore, 0.5 mM IPTG concentration was chosen for future studies. The cells were collected after 6 h of induction, as described previously. Next, four different induction temperatures (18, 25, 30 and 37 °C) were evaluated. The cell lysate corresponding to each condition was analyzed by SDS-PAGE and the protein expression was found to be highest when induced at 30 °C”. The results are presented in Figure 5C.
- There are no conclusions from the results obtained during induction of expression at different cultivation temperatures, there is only a link to the illustration.
Response: In light of reviewer’s comments, we have now included a sentence in the manuscript talking about the induction of expression at different cultivation temperatures. The cell lysate corresponding to each condition was analyzed by SDS-PAGE and the protein expression was found to be highest when induced at 30 °C (Lines – 561-562).
Minor:
- Line 215 - apparently chloramphenicol should be listed here instead of kanamycin.
Response: We thank the reviewers for pointing out the ambiguity. We have modified the statement in the revised manuscript.
- Lines 284 and 298 - Competent cells according to the protocol are suggested to be aliquoted in 100 μl, and later used in 50 μl. Perhaps it would be more convenient to aliquot 50 µL each?
Response: We thank the reviewers for pointing out the ambiguity. We have now modified it in the revised manuscript.
- Lines 326, 332, 356 - Since the work uses different lines carrying resistance to different antibiotics, it is better to write “LB medium with the appropriate antibiotic” instead of specifically indicating kanamycin.
Response: We agree with the reviewers and have corrected it in the revised manuscript.
- Line 629 – E. coli in italics
Response: We thank the reviewers for pointing out the mistake. We have now modified it in the revised manuscript.
Round 2
Reviewer 2 Report
Comments and Suggestions for Authors
Thank you for the clarification and the additional experiment.